# Prevalence and Prognostic Significance of Malnutrition in Patients with Abnormal Glycemic Status and Coronary Artery Disease: A Multicenter Cohort Study in China

**DOI:** 10.3390/nu15030732

**Published:** 2023-02-01

**Authors:** Tianyu Li, Xiaozeng Wang, Zhenyu Liu, Zheng Zhang, Yongzhen Zhang, Zhifang Wang, Yingqing Feng, Qingsheng Wang, Xiaogang Guo, Xiaofang Tang, Jingjing Xu, Ying Song, Yan Chen, Na Xu, Yi Yao, Ru Liu, Pei Zhu, Yaling Han, Jinqing Yuan

**Affiliations:** 1National Clinical Research Center for Cardiovascular Diseases, State Key Laboratory of Cardiovascular Disease, Fuwai Hospital, National Center for Cardiovascular Diseases, Chinese Academy of Medical Sciences and Peking Union Medical College, Beijing 100037, China; 2Cardiovascular Research Institute & Department of Cardiology, General Hospital of Northern Theater Command, Shenyang 110016, China; 3Department of Cardiology, Peking Union Medical College Hospital, Chinese Academy of Medical Sciences and Peking Union Medical College, Beijing 100730, China; 4Department of Cardiology, The First Hospital of Lanzhou University, Lanzhou 730000, China; 5Department of Cardiology, Peking University Third Hospital, Beijing 100191, China; 6Department of Cardiology, Xinxiang Central Hospital, Xinxiang 453002, China; 7Department of Cardiology, Guangdong Cardiovascular Institute, Guangzhou 510100, China; 8Department of Cardiology, The First Hospital of Qinhuangdao, Qinhuangdao 066000, China; 9Department of Cardiology, The First Affiliated Hospital of Zhejiang University, Hangzhou 314400, China; 10Department of Cardiology, Fuwai Hospital, National Center for Cardiovascular Diseases, Chinese Academy of Medical Sciences and Peking Union Medical College, Beijing 100037, China

**Keywords:** malnutrition, prediabetic state, diabetes mellitus, coronary artery disease

## Abstract

This study sought to investigate the prevalence and prognostic significance of malnutrition in patients with an abnormal glycemic status and coronary artery disease (CAD). This secondary analysis of a multicenter prospective cohort included 5710 CAD patients with prediabetes and 9328 with diabetes. Four objective tools were applied to assess the nutritional status of the study population. The primary endpoint was all-cause death. The association of malnutrition with clinical outcomes was examined using Cox proportional hazards regression. The proportion of malnutrition varied from 8% to 57% across the assessment tools. Diabetic patients were more likely to be malnourished than prediabetic patients. During a median follow-up of 2.1 years, 456 all-cause deaths occurred. The adjusted hazard ratios and 95% confidence interval for all-cause deaths of moderate–severe malnutrition defined by different tools ranged from 1.59 (1.03, 2.46) to 2.08 (0.92, 4.73) in prediabetic patients and 1.51 (1.00, 2.34) to 2.41 (1.78, 3.27) in diabetic patients. In conclusion, malnutrition is not rare in CAD patients with abnormal glycemic status. Moderate–severe malnutrition strongly predicted all-cause death regardless of the assessment tool. Assessing the nutritional status for all CAD patients with prediabetes and diabetes to identify individuals at high risk of all-cause death may help the risk assessment and prognosis improvement.

## 1. Introduction

Malnutrition refers to an abnormal physiological condition caused by inadequate, unbalanced, or excessive consumption of macronutrients or micronutrients, manifested as an altered body composition and diminished biological function [1]. Nutritional status has been of particular concern in the elderly, the frail, and cancer patients. The adverse prognostic impact of malnutrition in these populations has been well-established [2,3,4]. Recently, studies from Europe, America, and Asia have studied nutritional status by various means in patients with cardiovascular disease, such as heart failure, hypertension, cardiomyopathy, and coronary artery disease (CAD), and have reported that malnutrition is associated with prolonged hospital stay, increased in-hospital mortality, and decreased survival [5,6,7,8,9,10].

The growing population of patients with CAD and comorbid abnormal glycemic status has been a major public health problem in both developed and developing countries [11,12]. The current evidence raises awareness of the impact of nutritional status on the prognosis of patients with CAD [5,8,13,14,15] or diabetes [16,17,18,19]. However, most studies focused only on acute coronary syndrome (ACS) or elderly patients with diabetes lack the knowledge of the overall population with combined traits.

Various tools have been developed for nutritional assessment, including subjective questionnaires and objective methods. Objective nutritional assessment tools derived from anthropometric and laboratory measurements may be suitable for cardiologists to assess patients’ nutritional status easily and quickly in the outpatient setting or on admission, yet they are not widely examined in the population with CAD and comorbid abnormal glycemic status. We aimed to individually apply four objective nutritional assessment tools to picture nutritional status and corresponding prognostic consequences in CAD patients with prediabetes and diabetes.

## 2. Materials and Methods

### 2.1. Study Design, Setting, and Participants

The prospective observational multicenter cohort for ischemic and hemorrhage risk in coronary artery disease patients (PROMISE) is a prospective multicenter cohort study that aims to develop risk scores to quantify ischemic and bleeding risk in CAD patients in China. From January 2015 to May 2019, PROMISE recruited 18,701 hospitalized CAD patients in cardiology wards at Fuwai Hospital (National Center for Cardiovascular Diseases, Beijing, China) and eight other regional tertiary referral hospitals in mainland China. Inclusion criteria included an age of at least 18 years, diagnosis of CAD, indication of at least one antiplatelet drug, and willingness to participate in the study and sign informed consent. Exclusion criteria were life expectancy of fewer than six months and current participation in another interventional clinical trial. The selection of a treatment strategy and details of medication and intervention conformed to contemporaneous practice guidelines with consideration of patients’ preference. Demographic information, clinical characteristics, laboratory test results, medication administration data, details on coronary angiography and intervention, and information on in-hospital outcomes were extracted from electronic medical records. Medication lists and outcome data were followed up through outpatient visits, telephone interviews, text messages, and letters by an independent group of clinical research coordinators one year and two years after discharge. Investigator training, telephone recording, and an online follow-up system were applied to achieve comparable follow-ups across sites. Two independent cardiologists adjudicated endpoint events, and any disagreement was resolved by consensus. This study complied with the Declaration of Helsinki. The Ethics Committee of Fuwai Hospital approved the study protocol (protocol code: 2017-860, date of approval: 10 January 2017). All participants provided written informed consent.

Based on the above cohort, this secondary analysis investigated the prevalence and prognostic significance of malnutrition in CAD patients accompanied by abnormal glycemic status. Patients with prediabetes and diabetes were included in the present study. Patients with leukemia or lymphoma, end-stage liver or renal disease, missing data for height and weight, lymphocyte count, serum albumin, and total cholesterol were excluded from this study. Patients who died before discharge or the follow-up were excluded from the final analysis. This report complies with the Strengthening the Reporting of Observational Studies in Epidemiology (STROBE) statement.

### 2.2. Blood Sampling and Laboratory Testing

Venous blood samples were collected as required by routine clinical practice. Fasting blood glucose was assayed by an enzymatic hexokinase method. Glycated hemoglobin was assayed using a Tosoh Automated Glycohemoglobin Analyzer (HLC-723G8, Tokyo, Japan). Blood counts were measured using an automated blood cell counter. Serum albumin was measured with an automated chemistry analyzer (AU5400, Olympus, Tokyo, Japan) using bromocresol green dye method. Lipid profiles were measured using an automatic biochemistry analyzer (Hitachi 7150, Tokyo, Japan) in an enzymatic assay. High-sensitivity C-reactive protein was measured using immunoturbidimetry (Beckmann Assay, Bera, CA, USA). Estimated glomerular filtration rate was calculated with Asian modified modification of diet in renal disease equation.

### 2.3. Endpoints and Covariables

The primary endpoint was all-cause death. The secondary endpoint was major adverse cardiovascular and cerebrovascular event (MACCE), a composite of cardiac death, non-fatal myocardial infarction, and non-fatal stroke. All deaths were considered cardiac unless an unequivocal non-cardiac cause could be established. MI was diagnosed in accordance with the fourth universal definition of myocardial infarction. Strokes included ischemic stroke, hemorrhagic stroke, and transient ischemic attack.

Prediabetes was defined as a fasting blood glucose of 5.6–6.9 mmol/L, glycated hemoglobin ≥5.7% and <6.5% (≥39 mmol/mol and <48 mmol/mol), or 2 h blood glucose of oral glucose tolerance test of 7.8–11.0 mmol/L. Diabetes was defined as a fasting blood glucose ≥7.0 mmol/L, glycated hemoglobin ≥6.5% (≥48 mmol/mol), 2 h blood glucose of oral glucose tolerance test ≥11.1 mmol/L, oral antidiabetic medication or insulin use, or self-reported history of diabetes. Body mass index (BMI) was categorized into underweight (<18.5 kg/m^2^), normal weight (18.5–22.9 kg/m^2^), overweight (23.0–24.9 kg/m^2^), and obese (≥25 kg/m^2^), in accordance with the World Health Organization standard for Asian populations. Smoking status included current smoker, former smoker, and never smoker. A person who quit smoking for less than one month before admission was also considered a current smoker. Hypertension was defined as a blood pressure ≥140/90 mmHg on repeated measurements, antihypertensive medication use, or self-reported history of hypertension. Dyslipidemia was defined when at least one of the following criteria were met: total cholesterol ≥6.22 mmol/L, total triglyceride ≥2.26 mmol/L, low-density lipoprotein cholesterol ≥4.14 mmol/L, high-density lipoprotein cholesterol <1.04 mmol/L, lipid-lowering medication use, or self-reported history of dyslipidemia. Three-vessel disease was defined as an angiographically confirmed stenosis of ≥50% in all three main coronary arteries (left anterior descending artery, left circumflex artery, and right coronary artery).

### 2.4. Assessment of Nutritional Status

Four nutritional assessment tools were applied individually for malnutrition diagnosis and severity grading.

The global leadership initiative on malnutrition (GLIM) criteria are a consensus for diagnosing malnutrition [20]. A diagnosis of malnutrition can be established when at least one etiologic and one phenotypic criterion are met. Etiologic criteria are reduced food intake or assimilation and disease burden/inflammation. Patients with conditions that adversely affect eating, digestion, and absorption were considered to meet the first etiologic criterion. Disease burden refers to acute or chronic diseases that affect inflammatory status (Figure 1A). In line with prior studies [21], a high-sensitivity C-reactive protein >3.0 mg/L reflects inflammation. Phenotypic criteria applied in this study included low BMI and reduced muscle mass. A BMI of <18.5 kg/m^2^ in patients <70 years or <20.0 kg/m^2^ in patients ≥70 years is considered a low BMI for Asian populations. Reduced muscle mass was reflected by a fat-free mass index of <17.0 kg/m^2^ for males or <15.0 kg/m^2^ for females. Due to the lack of consensus on severity grading for Asian populations, we combined moderate and severe malnutrition for analysis. There is no mild malnutrition for the GLIM criteria.

Prognostic nutritional index (PNI) was first introduced in the context of gastrointestinal surgery and then reported in malignancy and cardiovascular diseases. It was calculated as serum albumin (g/L) + 5× absolute lymphocyte counts (×10^9^/L) as modified by Onodera et al. [22]. Based on our previous finding and the normal reference intervals of PNI in China, [23] a PNI of ≥43.0 is considered normal, and a PNI of 38.0–42.9 and <38.0 reflect moderate and severe malnutrition, respectively. Similar to the GLIM criteria, there is no mild malnutrition for PNI.

The controlling nutritional status (COUNT) score is developed to assess the nutritional status of hospitalized patients, considering total cholesterol in addition to lymphocyte count and serum albumin [24]. Scores are assigned according to the level of each component (Figure 1A) [8]. A score of 0–1 represents non-malnutrition; scores of 2–4, 5–8, and 9–12 represent mild, moderate, and severe malnutrition, respectively.

Nutritional risk index (NRI) is developed to detect hospitalized elderly people who are at risk of malnutrition [25]. In line with prior studies [8], NRI was calculated as 1.489 × serum albumin (g/L) + 41.7× (current weight/ideal weight) (Figure 1A). We set current weight/ideal weight = 1 when current weight exceeded ideal weight. NRI was classified into four levels: ≥100 (non-malnutrition), ≥97.5 and <100 (mild malnutrition), ≥83.5 and <97.5 (moderate malnutrition), and <83.5 (severe malnutrition).

### 2.5. Statistical Analysis

Categorical and continuous variables were expressed as numbers (percentages) and median (interquartile range), respectively. Central estimates and proportions across ordered groups were compared using Student’s *t*-test, Mann–Whitney U test, Kruskal–Wallis H test, or χ^2^ test as appropriate.

Incidence rates were visualized using bar charts. Survival curves were plotted using Kaplan–Meier method and compared using log-rank test. The association between malnutrition and clinical outcomes was examined using Cox proportional hazards regression by estimating hazard ratios and 95% confidence intervals. Relationships of PNI, the COUNT score, and NRI in continuous scale with clinical outcomes were examined with restricted cubic splines with 4 knots.

Covariables for adjustment included age, sex, BMI, smoking status, CAD presentation, hypertension, dyslipidemia, prior myocardial infarction, prior stroke, estimated glomerular filtration rate <60 mL/min/1.73 m^2^, left ventricular ejection fraction <40%, left main coronary artery/three-vessel disease, percutaneous coronary intervention, and medication based on clinical plausibility and significance in univariate Cox analysis.

Given the large difference in proportions of individuals with and without malnutrition, Cox analysis was repeated after propensity-score matching as a sensitivity analysis. The greedy algorithm was used with a caliper width equal to 0.2 of the standard deviation of the logit of the propensity score without replacement. Individuals with moderate–severe malnutrition were matched to those with mild and without malnutrition in a 1:4 manner.

Subgroup analysis was performed according to age (≥65 years versus <65 years), sex (women versus men), BMI (≥25 kg/m^2^ versus <25 kg/m^2^), and CAD presentation (chronic coronary syndrome versus ACS).

The performance of the four nutritional assessment tools for predicting all-cause death and MACCE was evaluated by C-statistic and was compared by continuous net reclassification improvement and integrated discrimination improvement.

Statistical analyses were conducted with R version 4.2.0 (R Core Team 2022, Vienna, Austria, www.R-project.org (accessed on 22 April 2022)). Figures were created by GraphPad Prism version 9.0.0 (GraphPad Software, San Diego, CA, USA, www.graphpad.com (accessed on 20 July 2022)). Two-tailed *p*-values of <0.05 were considered statistically significant.

## 3. Results

### 3.1. Study Population and Baseline Characteristics

The numbers of individuals at each stage are shown in Appendix A. Patients who died before discharge were more likely to present with diabetes and malnutrition, whereas baseline characteristics between patients with and without completed follow-up were comparable (Appendix A). A total of 15,038 participants were eligible for the final analysis, of whom 5710 (37.97%) had prediabetes and 9328 (62.03%) had diabetes. The median age of the study population was 62 years (interquartile range: 54–68 years). Among all the participants, 3904 (25.96%) were female, 129 (0.86%) were underweight, 3486 (23.18%) were overweight, 8863 (58.94%) were obese, and 7283 (48.43%) had ACS. Almost all participants (98.52%, *n* = 14,816) underwent coronary angiography, and most individuals (72.95%, *n* = 10,970) received percutaneous coronary intervention (Table 1).

Whatever the applied nutritional assessment tool, patients in the malnutrition groups were older, more likely to present with diabetes and ACS, have a lower BMI, have longer hospital stays, have prior strokes, have a reduced estimated glomerular filtration rate and left ventricular ejection fraction, and have complex coronary lesions. The malnutrition groups had more females, except when the COUNT score was applied. Only the NRI-defined malnutrition group had more current smokers. More data on baseline characteristics of patients with different nutritional statuses according to four assessment tools are detailed in Table 2. Baseline characteristics of patients grouped by glycemic status are shown in Appendix A. Baseline characteristics of matched patients with different nutritional statuses according to different assessment tools are shown in Appendix A.

### 3.2. Prevalence of Malnutrition

The proportion of malnutrition varied widely across the assessment tools (Figure 1A). Approximately 10% of participants were classified as malnourished by the GLIM criteria or PNI; there was no mild degree for the two assessment tools. Nearly 60% of participants were diagnosed with malnutrition by the COUNT score, with 53.65% having mild malnutrition. NRI, in between, classified about 20% of participants as malnourished; the proportion of moderate malnutrition exceeded that of mild malnutrition. The difference in proportions of moderate–severe malnutrition was relatively small between the GLIM criteria (9.82%), PNI (7.87%), and NRI (11.86%). In comparison, only 3.27% of participants were diagnosed with moderate–severe malnutrition by the COUNT score. The percentage of malnourished individuals in patients with diabetes was higher than in patients in a prediabetic state when the GLIM criteria, PNI, and the COUNT score were applied, while malnutrition assessed by NRI was comparable between the prediabetes and diabetes groups.

Figure 1B illustrates the lack of uniformity in diagnosis across the nutritional assessment tools. Malnutrition diagnosed by at least two assessment tools accounted for 17.84% and 21.11% of patients with prediabetes and diabetes, respectively. However, only 1.10% of patients in a prediabetic state and 1.95% of patients with diabetes were diagnosed with malnutrition by all four assessment tools, while 35.41% of patients in a prediabetic state and 32.57% of patients with diabetes were not classified as malnourished by any of the four assessment tools.

The proportion of malnutrition increased with decreasing BMI categories, regardless of the assessment tool. It should be noted that even obese, overweight, or normal-weight individuals could present with any degree of malnutrition. Although the proportion of malnutrition was higher in patients with diabetes than in patients in a prediabetic state, no difference in its distribution by BMI categories was observed according to glycemic status (Figure 1C). Patients ≥65 years, females, and patients with ACS were more likely to have malnutrition than their counterparts, regardless of the assessment tool, except that the proportion of malnutrition diagnosed by the COUNT score was higher in males than in females (Appendix A).

### 3.3. Association of Malnutrition and Clinical Outcomes

During a median follow-up of 2.1 years (interquartile range: 2.0–2.3 years), 456 all-cause deaths and 1087 MACCEs occurred. The corresponding values were 131 (2.29%) and 341 (5.97%) for patients with prediabetes, and 325 (3.48%) and 746 (8.00%) for those with diabetes.

Worsening nutritional status was associated with higher incidence rates of all-cause death and MACCE and lower 3-year overall and event-free survival rates, regardless of the assessment tool. The same trends were observed in populations with prediabetes and diabetes, though the latter had higher incidence rates of endpoint events and lower 3-year survival rates than the former (Figure 2, Appendix A).

Compared with non-malnutrition, moderate–severe malnutrition was significantly associated with higher risks of all-cause death and MACCE, before and after adjustment (multivariable Cox analysis and propensity-score matching) (Table 3). Patients in a prediabetic state with malnutrition had higher hazard ratios for endpoint events than patients with diabetes without malnutrition. We examined and ruled out the presence of an interaction between the glycemic status and nutritional status (Appendix A). After the adjustment, mild malnutrition yielded no significant association with the risk of all-cause death or MACCE in individuals with prediabetes and diabetes (Table 3). The results of the subgroup analysis are shown in Appendix A.

On a continuous scale, the risk of all-cause death declined with an elevating PNI and NRI and rose with an increasing COUNT score, regardless of the glycemic status. However, the 95% confidence interval of the hazard ratio of the COUNT score crossed the null (Figure 3). The analysis for MACCE generated similar results (Appendix A).

The comparisons of the four nutritional assessment tools for predicting clinical outcomes grouped by glycemic status are shown in Appendix A. All assessment tools showed better predictions for all-cause death than for MACCE.

## 4. Discussion

In the present study, nearly two-thirds of CAD patients with prediabetes and over two-thirds of CAD patients with diabetes were diagnosed with malnutrition by at least one assessment tool. However, poor diagnostic consistency across the assessment tools suggests that each tool can identify certain individuals at specific nutritional risks but will miss some individuals at other nutritional risks. This finding is not an isolated case [8]. The diabetes group comprised more malnourished patients than the prediabetes group, which may have resulted from increased protein catabolism, decreased protein anabolism, elevated urinary protein excretion, and probably, inadequate protein intake in the setting of diabetes. Prior small-scale research has investigated malnutrition in older patients with diabetes [16,17,19] and patients with ACS [5,8,14,15], respectively, and may not have been epidemiologically representative of the general population with abnormal glycemic status and CAD. Only one study [13] reported malnutrition in CAD patients with diabetes. Little is known about the nutritional status of patients with diabetes. The present study broadens the understanding of the nutritional status of CAD patients with abnormal glycemic status, and addresses the notion that malnutrition in patients with prediabetes should not be underestimated.

In the present study, the prevalence of moderate–severe malnutrition increased with decreasing BMI categories, whereas the trend was not observed for mild malnutrition. We hypothesized that early-stage malnutrition might be reflected by subclinical changes in biomarkers, and a weight loss-predominant clinical phenotype might emerge as the progression of malnutrition. Nevertheless, our findings suggested that moderate–severe malnutrition was not rare in normal-weight, overweight, and even obese patients. Moreover, evidence from the general Asian community [21] and patients with heart failure [26] showed that obese individuals with coexisting malnutrition had the worst prognosis. In fact, concern has been raised regarding the sensitivity of BMI for malnutrition screening, as the prevalence of overweight people and obesity is continuously rising [27,28]. The present study reports the proportion of malnutrition across BMI categories in patients with CAD and coexisting abnormal glycemic status, who carry a heavy burden of overweight and obesity, underscoring the necessity for a comprehensive assessment of nutritional status in addition to body weight.

Malnutrition is a common risk factor for poor prognosis in various populations, whether assessed by subjective or objective tools or single biomarkers. [5,8,13,14,15,16,17,18,19] Our findings confirmed its prognostic significance in patients with CAD and coexisting abnormal glycemic status. Notably, we presented a novel finding that the malnutrition-related risk for all-cause death and MACCE outweighed the diabetes-related risk. This finding, together with the observation in Medicare beneficiaries with diabetes [18], demonstrates that the risk of all-cause death is more attributable to malnutrition than to other chronic comorbidities, demonstrating that malnutrition is a potent and general negative prognostic factor. Given that malnutrition is a strong independent risk factor for poor prognosis, and it can aggravate diabetes by exacerbating insulin resistance [29] thereby synergistically worsening the prognosis of CAD, the significance of malnutrition screening and early intervention in patients in a prediabetic state should be emphasized.

Malnutrition diagnosed by any of the four nutritional assessment tools predicted all-cause death and MACCE in the study population to some extent. The COUNT score did not perform as well as the other tools. On the one hand, the COUNT score identified substantially more malnourished individuals than the other tools, the vast majority of whom were classified as mildly malnourished, which appeared to be driven mainly by low on-statin total cholesterol levels, leading to probable overestimation of the risk of mild malnutrition. On the other hand, no significant differences in the baseline total cholesterol levels were observed in the deceased versus surviving patients, and in patients who experienced MACCE versus those who did not, and a univariate analysis revealed no association between total cholesterol levels and all-cause death or MACCE. Thus, our results indicated that the COUNT score might not be an ideal nutritional assessment tool for CAD patients with abnormal glycemic status. Conversely, Sergio et al. [8] stated the superiority of the COUNT score and the significant inverse relationship between total cholesterol and mortality. As the authors discussed, we agree that this paradox exemplifies collider or index event bias [30]. Mild malnutrition diagnosed by NRI and the COUNT score had a neutral effect on all-cause death and MACCE after adjustment for confounders, with only moderate–severe malnutrition robustly associated with increased risks of clinical events. The limitation of the clinical application of PNI and the GLIM criteria is that there is no consensus on the optimal cutoff value of PNI, nor is there a consensus on the thresholds for corresponding anthropometric measurements and inflammation of the GLIM criteria. Randomized clinical trials are necessary to evaluate the value of different nutritional assessment tools as indicators of the efficacy of nutritional support in improving the prognosis of patients with abnormal glycemic status and CAD. A more comprehensive nutritional assessment system is needed to identify patients with various nutritional risks.

We recommend a simple and quick nutritional assessment of all patients with an abnormal glycemic status and CAD in outpatient clinics or on admission, to identify malnourished individuals at a high risk of poor prognosis. Objective tools derived from widely available anthropometric and laboratory measurements are suitable for these settings. In addition, malnutrition is a modifiable risk factor, suggesting that multi-disciplinary collaboration between endocrinologists, cardiologists, and dietitians can be helpful. Prior studies have shown that elderly malnourished patients with diabetes can benefit from diabetes-specific nutritional supplements [31,32,33], and individualized nutritional support during a hospital stay can improve the clinical outcomes of patients at nutritional risk [34]. Weight management is fundamental in the management of prediabetes, diabetes, and CAD, yet growing evidence highlights that the improvement of the underlying nutritional status should be emphasized while achieving the weight loss goal [35].

The present study has several strengths. First, it provides novel knowledge on the prevalence and prognostic significance of malnutrition in patients with CAD and comorbid abnormal glycemic status and several subpopulations of particular clinical interest, complementing the understanding of nutritional status in this extremely high-risk population. Second, the study applied four commonly used objective tools on categorical and continuous scales to assess nutritional status and its relationship with prognosis. Third, the participating centers were carefully selected throughout the country to achieve broad geographic and dietary representation. Other strengths include the large sample size and high follow-up rates.

However, the limitations of the study should be noted. Firstly, the observational nature raises concerns about residual confounding. Secondly, all participating centers were located in China, restricting the generalizability to individuals of other races/ethnicities. Thirdly, we had to simplify the GLIM criteria due to the lack of data on weight loss and energy requirements, which may result in an underestimation of the prevalence of malnutrition diagnosed by the GLIM criteria. Fourthly, we did not collect information about education, income, marital status, or diet, which could help us understand the causes of malnutrition. We did not have data on the number of cigarettes consumed daily either, which prevented us from adjusting for possible confounding on a continuous scale. Lastly, we did not follow up on the changes in nutritional status over time. Studies in other countries and regions are welcome to validate our work.

## 5. Conclusions

The prevalence of malnutrition varied across the assessment tools but was not rare in patients with CAD accompanied by abnormal glycemic status. Worsening nutritional status, especially moderate–severe malnutrition, significantly increased the risks of all-cause death and MACCE. After adjusting for traditional cardiovascular risk factors, malnutrition remained an independent adverse prognostic factor in CAD patients with prediabetes and diabetes, even stronger than diabetes. Assessing nutritional status in all CAD patients with prediabetes and diabetes to identify individuals at high risk of all-cause death may help risk assessments and prognosis improvement.

## Figures and Tables

**Figure 1 nutrients-15-00732-f001:**
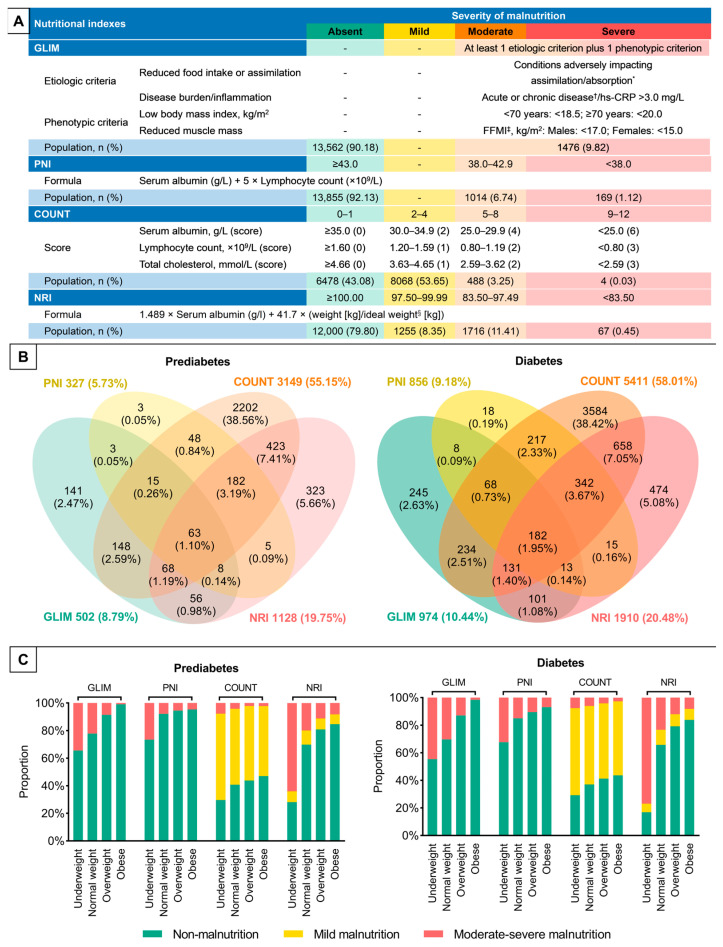
Prevalence of malnutrition. (**A**) Diagnosis and severity grading of malnutrition. (**B**) Number and percentage of malnourished patients according to different assessment tools. (**C**) Proportion of different nutritional statuses according to each assessment tool across body mass index categories (underweight (<18.5 kg/m^2^), normal weight (18.5–22.9 kg/m^2^), overweight (23.0–24.9 kg/m^2^) and obese (≥25 kg/m^2^)). * e.g., esophagus cancer, inflammatory bowel disease, short bowel syndrome, or history of bariatric surgery; † e.g., sepsis, trauma, chronic obstructive pulmonary disease, or congestive heart failure; ‡ FFMI (kg/m^2^) = weight (kg) × (1 − body fat (%)/100)/(height (m))^2^; § ideal weight, kg: men: height (cm) − 100 − ((height (cm) − 150)/4); women: height (cm) − 100 − ((height (cm) − 150)/2.5). COUNT, controlling nutritional status; FFMI, free-fat mass index; GLIM, global leadership initiative on malnutrition; hs-CRP, high-sensitivity C-reactive protein; NRI, nutritional risk index; and PNI, prognostic nutritional index.

**Figure 2 nutrients-15-00732-f002:**
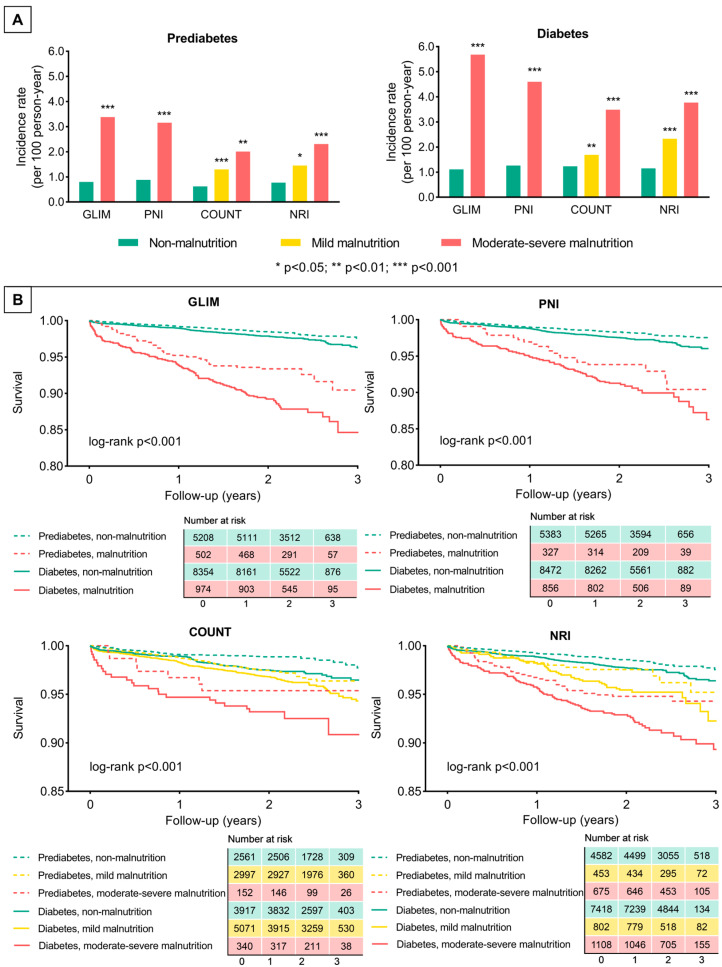
Incidences of all-cause death. (**A**) Incidence rate of all-cause death according to different assessment tools. (**B**) Survival curves by glycemic status and nutritional status according to each assessment tool. GLIM, global leadership initiative on malnutrition; PNI, prognostic nutritional index; COUNT, controlling nutritional status; and NRI, nutritional risk index.

**Figure 3 nutrients-15-00732-f003:**
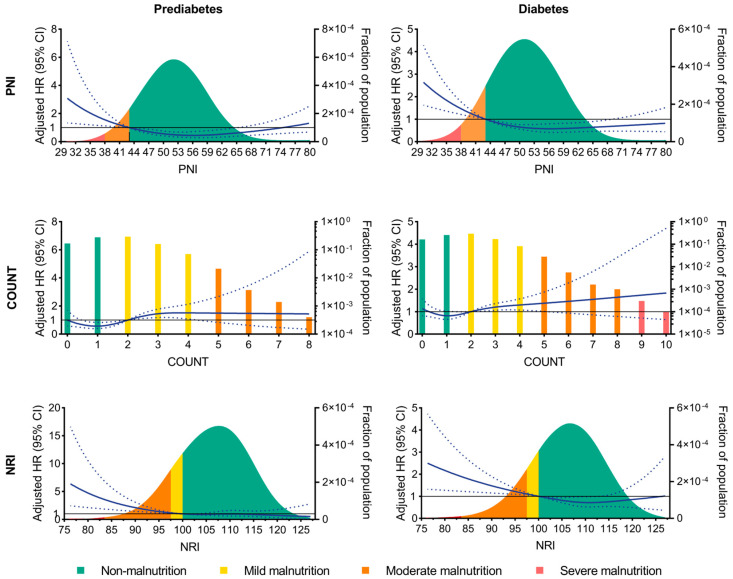
RCSs of PNI, COUNT, and NRI for predicting all-cause death. CI, confidence interval; COUNT, controlling nutritional status; GLIM, global leadership initiative on malnutrition; HR, hazard ratio; NRI, nutritional risk index; PNI, prognostic nutritional index; and RCS, restricted cubic spline.

**Table 1 nutrients-15-00732-t001:** Baseline characteristics of all participants.

	All Participants (*n* = 15,038)
Demographic characteristics	
Age, years	62 (54–68)
Female	3904 (25.96)
Body mass index, kg/m^2^	25.71 (23.67–27.78)
Smoking status	
Current smoker	3674 (24.43)
Former smoker	4928 (32.77)
Never smoker	6436 (42.80)
Clinical characteristics	
CAD presentation	
ACS	7283 (48.43)
CCS	7755 (51.57)
Length of stay, day	5 (3–8)
Glycemic status	
Prediabetes	5710 (37.97)
Diabetes	9328 (62.03)
Hypertension	10,843 (72.10)
Dyslipidemia	13,903 (92.45)
Peripheral artery disease	816 (5.43)
COPD	242 (1.61)
Prior myocardial infarction	2642 (17.57)
Prior stroke	2319 (15.42)
Laboratory tests	
FBG, mmol/L	6.29 (5.37–8.04)
HbA1c, %	6.1 (5.8–7.1)
Lymphocyte count, ×10^9^/L	1.84 (1.37–2.56)
Serum albumin, g/L	42.5 (38.9–46.1)
hs-CRP, mg/L	1.92 (1.03–4.73)
Total cholesterol, mmol/L	3.99 (3.37–4.76)
eGFR <60 mL/min/1.73 m^2^	438 (2.91)
LVEF <40%	431 (2.87)
Angiographic characteristics	
Coronary angiography	14,816 (98.52)
LMCA/three-vessel disease	7071 (47.02)
SYNTAX score	
≤22	12,254 (82.75)
23–32	1958 (13.22)
≥33	596 (4.02)
PCI	10,970 (72.95)
Medication	
Aspirin	14,829 (98.61)
P2Y_12_ inhibitors	13,530 (89.97)
Statins	14,700 (97.75)
β-blockers	12,183 (81.01)
ACEIs/ARBs	9417 (62.62)
Malnutrition	
GLIM	1476 (9.82)
PNI	1183 (7.87)
COUNT	8560 (56.92)
Mild	8068 (53.65)
Moderate–severe	492 (3.27)
NRI	3038 (20.21)
Mild	1255 (8.35)
Moderate–severe	1783 (11.86)

Values are presented as number (%) or median (interquartile range). ACEI, angiotensin-converting enzyme inhibitor; ACS, acute coronary syndrome; ARB, angiotensin-receptor blocker; CCS, chronic coronary syndrome; COPD, chronic obstructive pulmonary disease; COUNT, controlling nutritional status; eGFR, estimated glomerular filtration rate; FBG, fasting blood glucose; GLIM, global leadership initiative on malnutrition; HbA1c, glycated hemoglobin; hs-CRP, high-sensitivity C-reactive protein; LMCA, left main coronary artery; LVEF, left ventricular ejection fraction; NRI, nutritional risk index; PCI, percutaneous coronary intervention; PNI, prognostic nutritional index; and SYNTAX, synergy between percutaneous coronary intervention with Taxus and cardiac surgery.

**Table 2 nutrients-15-00732-t002:** Baseline characteristics of patients with different nutritional statuses according to four assessment tools.

	GLIM			PNI			COUNT				NRI			
	Malnutrition (*n* = 1476)	Non-Malnutrition (*n* = 13,562)	*p*	Malnutrition (*n* = 1183)	Non-Malnutrition (*n* = 13,855)	*p*	Moderate–Severe Malnutrition (*n* = 492)	Mild Malnutrition (*n* = 8068)	Non-Malnutrition (*n* = 6478)	*p*	Moderate–Severe Malnutrition (*n* = 1783)	Mild Malnutrition (*n* = 1255)	Non-Malnutrition (*n* = 12,000)	*p*
Demographic characteristics														
Age, years	72 (65–78)	61 (54–67)	<0.001	67 (61–75)	61 (54–68)	<0.001	66 (59–74)	62 (55–69)	61 (53–67)	<0.001	67 (60–74)	64 (58–70)	61 (54–67)	<0.001
Female	744 (50.41)	31 (23.30)	<0.001	352 (29.75)	3552 (25.64)	0.002	112 (22.76)	1830 (22.68)	1962 (30.29)	<0.001	524 (29.39)	388 (30.92)	2992 (24.93)	<0.001
Body mass index, kg/m^2^	22.49 (21.10–23.83)	26.03 (24.22–28.09)	<0.001	24.77 (22.59–26.85)	25.83 (23.81–27.99)	<0.001	24.73 (22.57–27.02)	25.63 (23.62–27.73)	25.95 (23.88–28.08)	<0.001	24.22 (21.61–26.59)	25.34 (23.26–27.68)	25.95 (24.02–28.06)	<0.001
Smoking status														
Current smoker	254 (17.21)	3420 (25.22)	<0.001	267 (22.57)	3407 (24.59)	0.036	104 (21.14)	1766 (21.89)	1804 (27.85)	<0.001	484 (27.15)	309 (24.62)	2881 (24.01)	<0.001
Former smoker	435 (29.47)	4493 (33.13)		368 (31.11)	4560 (32.91)		173 (35.16)	2889 (35.81)	1866 (28.81)		440 (24.68)	364 (29.00)	4124 (34.37)	
Never smoker	787 (53.32)	5649 (41.65)		548 (46.32)	5888 (42.50)		215 (43.70)	3413 (42.30)	2808 (43.35)		859 (48.18)	582 (46.37)	4995 (41.62)	
Clinical characteristics														
CAD presentation														
ACS	986 (66.80)	6297 (46.43)	<0.001	996 (84.19)	6287 (45.38)	<0.001	307 (62.40)	3740 (46.36)	3236 (49.95)	<0.001	1271 (71.28)	750 (59.76)	5262 (43.85)	<0.001
CCS	490 (33.20)	7265 (53.57)		187 (15.81)	7568 (54.62)		185 (37.60)	4328 (53.64)	3242 (50.05)		512 (28.72)	505 (40.24)	6738 (5.15)	
Length of stay, day	7 (4–10)	5 (3–7)	<0.001	7 (5–11)	5 (3–7)	<0.001	6 (4–10)	5 (3–8)	5 (3–8)	<0.001	6 (4–10)	5 (3–9)	5 (3–7)	<0.001
Glycemic status														
Prediabetes	502 (34.01)	5208 (38.40)	<0.001	327 (27.64)	5383 (38.85)	<0.001	152 (30.89)	2997 (37.15)	2561 (39.53)	<0.001	675 (37.86)	453 (36.10)	4582 (38.18)	0.348
Diabetes	974 (65.99)	8354 (61.60)		856 (72.36)	8472 (61.15)		340 (69.11)	5071 (62.85)	3917 (60.47)		1108 (62.14)	802 (63.90)	7418 (61.82)	
Hypertension	1096 (74.25)	9747 (71.87)	0.052	853 (72.10)	330 (27.90)	0.999	334 (67.89)	5918 (73.35)	4592 (70.89)	<0.001	1167 (65.45)	887 (70.68)	8789 (73.24)	<0.001
Dyslipidemia	1364 (92.41)	12,539 (92.46)	0.951	1096 (92.65)	12,807 (92.44)	0.793	464 (94.31)	7500 (92.96)	5939 (91.68)	0.004	1511 (84.74)	1121 (89.32)	11,271 (93.92)	<0.001
Peripheral artery disease	99 (6.71)	717 (5.29)	0.022	52 (4.40)	764 (5.51)	0.103	26 (5.28)	481 (5.96)	309 (4.77)	0.007	84 (4.71)	61 (4.86)	671 (5.59)	0.202
COPD	84 (5.69)	158 (1.17)	<0.001	47 (3.97)	195 (1.41)	<0.001	20 (4.07)	134 (1.66)	88 (1.36)	<0.001	42 (2.36)	25 (1.99)	175 (1.46)	0.010
Prior myocardial infarction	252 (17.07)	2390 (17.62)	0.598	205 (17.33)	2437 (17.59)	0.821	106 (21.54)	1584 (19.63)	952 (14.70)	<0.001	287 (16.10)	209 (16.65)	2146 (17.88)	0.122
Prior stroke	288 (19.51)	2031 (14.98)	<0.001	208 (17.58)	2111 (15.24)	0.032	79 (16.06)	1324 (16.41)	916 (14.14)	<0.001	324 (18.17)	172 (13.71)	1823 (15.19)	0.001
Laboratory tests														
FBG, mmol/L	6.88 (5.60–8.80)	6.24 (5.35–7.95)	<0.001	7.37 (5.98–9.59)	6.21 (5.33–7.91)	<0.001	6.60 (5.42–8.48)	6.24 (5.30–7.95)	6.33 (5.43–8.11)	<0.001	6.33 (5.17–8.38)	6.19 (5.03–8.02)	6.30 (5.42–8.00)	<0.001
HbA1c, %	6.15 (5.80–7.10)	6.10 (5.80–7.10)	0.520	6.0 (5.7–7.0)	6.2 (5.8–7.1)	<0.001	6.0 (5.7–6.9)	6.1 (5.8–7.0)	6.2 (5.8–7.2)	<0.001	6.1 (5.8–7.1)	6.1 (5.8–7.2)	6.2 (5.8–7.1)	0.347
Lymphocyte count, ×10^9^/L	1.66 (1.21–2.27)	1.86 (1.39–2.58)	<0.001	1.15 (0.87–1.60)	1.89 (1.44–2.60)	<0.001	0.96 (0.69–1.44)	1.53 (1.18–2.21)	2.13 (1.77–2.92)	<0.001	2.12 (1.41–15.20)	1.97 (1.41–11.50)	1.80 (1.36–2.41)	<0.001
Serum albumin, g/L	39.0 (35.9–43.2)	42.8 (39.3–46.4)	<0.001	35.0 (33.1–36.4)	43.0 (39.8–46.5)	<0.001	35.5 (32.5–41.6)	42.2 (38.6–45.9)	43.1 (39.8–46.7)	<0.001	35.8 (34.0–36.9)	38.4 (37.8–38.9)	44 (41.0–47.0)	<0.001
hs-CRP, mg/L	6.73 (3.68–11.45)	1.92 (0.94, 3.71)	<0.001	6.02 (1.92–11.56)	1.92 (0.97–4.15)	<0.001	2.82 (1.55–10.71)	1.92 (0.86–4.35)	1.93 (1.23–4.88)	<0.001	2.36 (1.85–10.30)	1.92 (1.25–6.29)	1.92 (0.95–4.17)	<0.001
Total cholesterol, mmol/L	4.10 (3.41–4.86)	3.98 (3.36–4.74)	0.011	3.86 (3.26–4.50)	4.00 (3.38–4.78)	0.001	3.08 (2.54–3.50)	3.48 (3.09–4.08)	4.70 (4.13–5.32)	<0.001	3.88 (3.25–4.58)	3.80 (3.23–4.63)	4.04 (3.40–4.79)	<0.001
eGFR < 60 mL/min/1.73 m^2^	199 (13.48)	239 (1.76)	<0.001	108 (9.13)	330 (2.38)	<0.001	33 (6.71)	237 (2.94)	169 (2.61)	<0.001	139 (7.80)	55 (4.38)	244 (2.03)	<0.001
LVEF < 40%	129 (8.74)	302 (2.23)	<0.001	77 (6.51)	354 (2.56)	<0.001	29 (5.89)	232 (2.88)	171 (2.64)	<0.001	103 (5.78)	34 (2.71)	294 (2.45)	<0.001
Angiographic characteristics														
Coronary angiography	1441 (97.63)	13,375 (98.62)	0.003	1153 (97.46)	13,663 (98.61)	<0.001	471 (95.73)	7959 (98.65)	6386 (98.58)	<0.001	1716 (96.24)	1223 (97.45)	11,877 (98.98)	<0.001
LMCA/three-vessel disease	825 (55.89)	6246 (46.06)	<0.001	656 (55.45)	6415 (46.30)	<0.001	248 (50.41)	3778 (46.83)	3046 (47.02)	0.346	953 (53.45)	643 (51.24)	5475 (45.62)	<0.001
SYNTAX score														
≤22	1091 (75.82)	11,163 (83.50)	<0.001	889 (77.17)	11,365 (83.22)	<0.001	373 (79.19)	6643 (83.52)	5237 (82.05)	0.004	1284 (74.91)	952 (77.91)	10,018 (84.38)	<0.001
23–32	235 (16.33)	1723 (12.89)		188 (16.32)	1770 (12.96)		65 (13.80)	1010 (12.70)	883 (13.83)		286 (16.69)	196 (16.04)	1476 (12.43)	
≥33	113 (7.85)	483 (3.61)		75 (6.51)	521 (3.82)		33 (7.01)	301 (3.78)	263 (4.12)		144 (8.40)	74 (6.06)	378 (3.18)	
PCI	1105 (74.86)	9865 (72.74)	0.081	950 (80.30)	10,020 (72.32)	<0.001	356 (72.36)	5823 (72.17)	4791 (73.96)	0.053	1313 (73.64)	939 (74.82)	8718 (72.65)	0.202
Medication														
Aspirin	1445 (97.90)	13,384 (98.69)	0.014	1154 (97.55)	13,675 (98.70)	0.001	475 (96.54)	7967 (98.75)	6388 (98.61)	<0.001	1748 (98.04)	1233 (98.25)	11,848 (98.73)	0.033
P2Y_12_ inhibitors	1357 (91.94)	12,173 (89.76)	0.008	1088 (91.97)	12,442 (89.80)	0.017	451 (91.67)	7271 (90.12)	5809 (89.67)	0.358	1631 (91.48)	1139 (90.76)	10,760 (89.67)	0.038
Statins	1430 (96.88)	13,270 (97.85)	0.018	1144 (96.70)	13,556 (97.84)	0.011	482 (97.97)	7897 (97.88)	6321 (97.58)	0.445	1726 (96.80)	1215 (96.81)	11,759 (97.99)	<0.001
β-blockers	1178 (79.81)	11,005 (81.15)	0.214	934 (78.95)	11,249 (81.19)	0.059	385 (78.25)	6629 (82.16)	5169 (79.79)	<0.001	1186 (66.52)	915 (72.91)	10,082 (84.02)	<0.001
ACEIs/ARBs	907 (61.45)	8510 (62.75)	0.327	790 (66.78)	8627 (62.27)	0.002	302 (61.38)	5038 (62.44)	4077 (62.94)	0.703	1099 (61.64)	838 (66.77)	7480 (62.33)	0.006

Values are presented as number (%) or median (interquartile range). ACEI, angiotensin-converting enzyme inhibitor; ACS, acute coronary syndrome; ARB, angiotensin-receptor blocker; CCS, chronic coronary syndrome; COPD, chronic obstructive pulmonary disease; COUNT, controlling nutritional status; eGFR, estimated glomerular filtration rate; FBG, fasting blood glucose; GLIM, global leadership initiative on malnutrition; HbA1c, glycated hemoglobin; hs-CRP, high-sensitivity C-reactive protein; LM/TVD, left main coronary artery/three-vessel disease; LVEF, left ventricular ejection fraction; MI, myocardial infarction; NRI, nutritional risk index; PAD, peripheral artery disease; PCI, percutaneous coronary intervention; PNI, prognostic nutritional index; and SYNTAX, synergy between percutaneous coronary intervention with Taxus and cardiac surgery.

**Table 3 nutrients-15-00732-t003:** Associations of malnutrition assessed by four assessment tools with all-cause death and MACCE, grouped by glycemic status.

	All-Cause Death		MACCE	
	Prediabetes	Diabetes	Prediabetes	Diabetes
GLIM (non-malnutrition as reference)				
Crude HR (95% CI)	3.49 (2.28, 5.33) *	4.39 (3.45, 5.57) *	1.91 (1.34, 2.70) *	2.50 (2.05, 3.04) *
Adjusted HR (95% CI)	1.65 (1.00, 2.71) ^†^	2.41 (1.78, 3.27) *	1.01 (0.69, 1.48)	1.62 (1.29, 2.05) *
PSM HR (95% CI)	2.58 (0.86, 7.73)	2.72 (1.43, 5.20) ^‡^	0.74 (0.33, 1.66)	1.99 (1.21, 3.25) ^‡^
PNI, categorical (non-malnutrition as reference)				
Crude HR (95% CI)	3.59 (2.29, 5.63) *	3.62 (2.82, 4.63) *	2.59 (1.84, 3.64) *	2.27 (1.86, 2.77) *
Adjusted HR (95% CI)	1.84 (1.14, 2.96) ^†^	1.83 (1.41, 2.39) *	1.70 (1.19, 2.44) ^‡^	1.46 (1.18, 1.81) *
PSM HR (95% CI)	1.78 (1.10, 2.90) ^†^	1.77 (1.34, 2.33) ^‡^	1.55 (1.04, 2.33) ^†^	1.40 (1.09, 1.80) ^‡^
COUNT, categorical (non-malnutrition as reference)				
Mild malnutrition				
Crude HR (95% CI)	2.09 (1.42, 3.08) *	1.38 (1.09, 1.75) ^‡^	1.52 (1.19, 1.94) *	1.25 (1.06, 1.47) ^‡^
Adjusted HR (95% CI)	1.86 (1.25, 2.75) ^‡^	1.21 (0.95, 1.54)	1.43 (1.11, 1.83) ^†^	1.13 (0.96, 1.33)
PSM HR (95% CI)	1.01 (0.32, 2.49)	1.82 (1.01, 3.50) ^†^	1.34 (0.66, 2.75)	1.63 (1.04, 2.55) ^†^
Moderate–severe malnutrition				
Crude HR (95% CI)	3.23 (1.44, 7.26) ^‡^	2.94 (1.93, 4.48) *	2.03 (1.12, 3.70) ^†^	1.91 (1.36, 2.69) *
Adjusted HR (95% CI)	2.08 (0.92, 4.73)	1.51 (1.00, 2.34) ^†^	1.56 (0.85, 2.86)	1.26 (0.89, 1.80)
PSM HR (95% CI)	1.92 (1.01, 3.85) ^†^	1.44 (1.05, 3.23) ^†^	1.85 (1.00, 4.19) ^†^	1.46 (0.88, 2.41)
NRI, categorical (non-malnutrition as reference)				
Mild malnutrition				
Crude HR (95% CI)	1.88 (1.03, 3.27) ^†^	2.03 (1.45, 2.83) *	1.45 (0.98, 2.14)	1.41 (1.09, 1.82) *
Adjusted HR (95% CI)	1.39 (0.79, 2.44)	1.43 (1.02, 2.01) ^†^	1.20 (0.81, 1.79)	1.17 (0.90, 1.51)
PSM HR (95% CI)	1.35 (0.69, 2.62)	1.35 (0.90, 2.03)	1.01 (0.36, 2.87)	1.32 (0.74, 2.35)
Moderate–severe malnutrition				
Crude HR (95% CI)	2.98 (2.01, 4.41) *	3.26 (2.55, 4.18) *	1.84 (1.37, 2.48) *	2.20 (1.82, 2.66) *
Adjusted HR (95% CI)	1.59 (1.03, 2.46) ^†^	1.62 (1.24, 2.14) *	1.22 (0.88, 1.68)	1.49 (1.22, 1.83) *
PSM HR (95% CI)	1.67 (1.07, 2.62) ^†^	1.80 (1.36, 2.38) *	1.14 (0.60, 2.16)	1.56 (1.04, 2.32) *

* *p* < 0.001; † *p* < 0.05; and ‡ *p* < 0.01. Adjusted for age, sex, body mass index, smoking status, presentation of coronary artery disease, hypertension, dyslipidemia, prior myocardial infarction, prior stroke, estimated glomerular filtration rate < 60 mL/min/1.73 m^2^, left ventricular ejection fraction < 40%, left main coronary artery/three-vessel disease, percutaneous coronary intervention, and medication. CI, confidence interval; COUNT, controlling nutritional status; GLIM, global leadership initiative on malnutrition; HR, hazard ratio; MACCE, major adverse cardiovascular and cerebrovascular event; NRI, nutritional risk index; PNI, prognostic nutritional index; and PSM, propensity-score matching.

## Data Availability

The datasets generated and analyzed during the current study are available from the corresponding author upon reasonable request.

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
