# Peer review of "Prevalence and Prognostic Significance of Malnutrition in Patients with Abnormal Glycemic Status and Coronary Artery Disease: A Multicenter Cohort Study in China"

_nutrients, 2023, doi:10.3390/nu15030732_

Round 1

Reviewer 1 Report

The study is very interesting and thoroughly conducted by a team of scientists. 

Using several methods to assess the nutritional status of patients was a very good idea. However, the work has several shortcomings that need to be addressed. 

1. The one weakness of the work presented is the short introduction. 

It might have been worthwhile to add information that nutritional status is a predictor of survival in CVD in Europe as well and is studied by various means.

https://www.mdpi.com/1660-4601/19/10/5827/htm

https://www.mdpi.com/2072-6643/12/10/3091 

https://www.mdpi.com/2072-6643/12/8/2330

2. The methodology is presented in a clear and correct manner

3. The results are presented correctly. 

4. Please add a separate paragraph study limitation before the conclusions

5. Please add practical implications before the conclusions 

6. I am not quite sure where the conclusions are in this study? Please create a "Conclusions" paragraph  

Reviewer 2 Report

Lack of control for smoking is a fundamental limitation of this manuscript.  Smoking is mentioned once in the article and appears limited to "current smoking" status.  A person may have smoked 1 pack a day for 30 years - quit smoking prior to enrolled in the study - and such information appears lost.  Smoking is the #1 driver of mortality in the US.  What is it in China?  Regardless, smoking is strongly related to malnutrition and is not addressed.

I found this manuscript confusing to read.  First, it was confusing to have 4 different measures of malnutrition which were poorly defined.  The first measure of malnutrition (https://aspenjournals.onlinelibrary.wiley.com/doi/10.1002/jpen.1806) appears to have been defined recently (2020), is based on expert opinion, has not been validated, and I failed to find which  phenotypic criteria was used in this study.  The three other measures of malnutrition were defined better but it is unclear why the reader should juggle 4 different definitions of malnutrition.  I suggest the authors choose one measure of malnutrition as primary.  (The three other definitions could become a detail in sensitivity analyses.)

The abstract does not present a measure of association between malnutrition and all-cause death or MACCE (This measure - when reported - should provide a detailed adjustment for smoking history)..  The authors report that the link between malnutrition and mortality is their primary aim and yet do not present results in abstract.  Please present the results of the primary aim in the abstract.  The reader is presented with findings other than the primary aim in the abstract – confusing.

Why is blood sampling and laboratory testing presented – once again this appears unrelated to primary aim.

I have a hard time following why suddenly diabetes and pre-diabetes become the apparent exposures of interest in table 1.  I expected to see a table with the different levels of malnutrition.  Is this a report on diabetes or malnutrition?  Focus appears unclear.

Then the authors digress in presenting how the different malnutrition tools lead to widely varying conclusions.  This is once again a digression.  What is the aim of this report?  To evaluate different malnutrition tools or to assess how malnutrition impacts mortality?

Minimal socio-demographic information is presented.  Measures of SES, education, ...?  

Reviewer 3 Report

This study sought to investigate the prevalence and prognostic significance of malnutrition in patients with abnormal glycemic status and coronary artery disease (CAD). The study is very interesting, both for the topic and the target population examined

Some comments below:

The introduction is a little poor. Please argue better by integrating some topics. For example, integrate some epidemiological data. The adult population is certainly more prone to malnutrition, frailty and nutritional imbalance. Please review the bibliography (10.1016/j.arr.2020.101148, 10.1111/joim.13384). 

In addition, it is useful to better detail all the tools used in the literature to investigate malnutrition status, in the methodology section.

Methodology: please provide details of the study's ethics committee. The statistical methodology is very robust. The discussions support the results

Round 2

Reviewer 1 Report

Thank you. Paper is ready for publication. 

Reviewer 2 Report

Thank you for the detailed responses to the initial comments.

Reviewer 3 Report

Well improved